# *Chlorella pyrenoidosa* Polysaccharide CPP-3a Promotes M1 Polarization of Macrophages via TLR4/2-MyD88-NF-κB/p38 MAPK Signaling Pathways

**DOI:** 10.3390/md23070290

**Published:** 2025-07-16

**Authors:** Yihua Pi, Qingxia Yuan, Shaoting Qin, Chundie Lan, Qingdong Nong, Chenxia Yun, Haibo Tang, Jing Leng, Jian Xiao, Longyan Zhao, Lifeng Zhang

**Affiliations:** 1Guangxi Key Laboratory of Translational Medicine for Treating High-Incidence Infectious Diseases with Integrative Medicine, Guangxi University of Chinese Medicine, Nanning 530200, China; piyh@gxtcmu.edu.cn (Y.P.); qst@163.com (S.Q.); lani0716@163.com (C.L.); nongqd@gxtcmu.edu.cn (Q.N.); yuncx@gxtcmu.edu.cn (C.Y.); tanghb@gxtcmu.edu.cn (H.T.); lj986771588@163.com (J.L.); xiaoj@gxtcmu.edu.cn (J.X.); 2Guangxi Key Laboratory of Marine Drugs, University Engineering Research Center of High-Efficient Utilization of Marine Traditional Chinese Medicine Resources, Institute of Marine Drugs, University of Chinese Medicine, Nanning 530200, China; qingxiayuan@163.com

**Keywords:** *Chlorella* polysaccharide, macrophage, M1 polarization, TLR4, NF-κB

## Abstract

The immunomodulatory polysaccharide CPP-3a, purified from *Chlorella pyrenoidosa*, was investigated for its effects on RAW264.7 macrophages and underlying mechanisms, revealing that CPP-3a significantly enhanced phagocytic capacity and nitric oxide production while upregulating pro-inflammatory cytokines TNF-α and IL-6 and elevating the co-stimulatory molecule CD86, collectively driving robust M1 polarization. Mechanistically, TLR4-, TLR2-specific inhibitors, and TLR4-knockout cells confirmed TLR4 as the primary receptor for CPP-3a, with TLR2 playing a secondary role in cytokine modulation. CPP-3a activated NF-κB and p38 MAPK signaling pathways via the MyD88-dependent pathway, evidenced by phosphorylation of NF-κB/p65 with its nuclear translocation and increased phosphorylation of p38 MAPK, with these signaling activations further validated by specific pathway inhibitors that abolished M1 polarization phenotypes. Collectively, CPP-3a emerges as a potent TLR4-targeted immunomodulator with adjuvant potential for inflammatory and infectious diseases.

## 1. Introduction

*Chlorella*, a single-celled green microalga of the phylum Chlorophyta, inhabits both freshwater and marine environments [1,2]. *Chlorella pyrenoidosa*, a prominent species within this genus, is rich in structurally diverse polysaccharides that exhibit broad-spectrum biological activities, including immunomodulatory, antioxidant, and prebiotic functions, leading to its designation as a “green healthy food” by the Food and Agriculture Organization of the United Nations (FAO) [3,4,5]. In our previous study, we successfully purified the predominant non-starch polysaccharide fraction, CPP-3a, and elucidated its detailed structural characteristics. CPP-3a is identified as a novel heteropolysaccharide with a high molecular weight and highly branched structures [6]. Subsequent pharmacological assays revealed that CPP-3a significantly influences the morphology of immature dendritic cells (DCs) and enhances the expression of maturation markers, including CD80, CD86, and MHC I. This maturation process is associated with a decrease in phagocytic capacity and an increase in T-cell stimulatory ability, underscoring CPP-3a’s potential to modulate the immune system and suggesting its broad therapeutic applications in immune-related disorders [6].

In addition to DCs, other cell types, such as macrophages, are integral to the immune system and can possess the ability to adopt various activation states in response to environmental stimuli. Upon activation, macrophages can differentiate into distinct subpopulations, including classically activated (M1) macrophages and alternatively activated (M2) macrophages, each characterized by specific functional roles. M1 macrophages secrete pro-inflammatory cytokines such as tumor necrosis factor-alpha (TNF-α) and interleukin-6 (IL-6), thereby playing a pivotal role in host defense against pathogens and tumor cells through the modulation of immune responses [7,8]. Previous research has demonstrated that polysaccharides can induce M1 polarization in macrophages, thus exhibiting immunomodulatory activity.

The molecular mechanisms underlying macrophage polarization are intricate, involving a multitude of signaling pathways [9,10,11,12,13]. Notably, the activation of the nuclear factor kappa-light-chain-enhancer of activated B cells (NF-κB) and mitogen-activated protein kinases (MAPK) pathways is critical in the transcriptional regulation of pro-inflammatory cytokines in M1 macrophages [14,15,16]. Furthermore, Toll-like receptors (TLRs), including TLR4 and TLR2, are vital for the recognition of pathogen-associated molecular patterns, thereby initiating these signaling cascades [17]. A comprehensive understanding of these pathways is crucial for the development of strategies aimed at modulating macrophage functions in various pathological conditions.

Previous studies have demonstrated that *Chlorella* spp. polysaccharide interventions exhibit immunomodulatory activity in macrophages [18,19,20]. However, the majority of research has not concentrated on the detailed structural characterization of the purified polysaccharide fractions. Our prior work identified CPP-3a as a novel heteropolysaccharide with a complex structure featuring multiple sugar residues and glycosidic linkages. This purified fraction demonstrated significant immunostimulatory effects by inducing dendritic cell (DC) maturation, where CPP-3a-treated DCs promoted allogeneic naïve CD8^+^/CD4^+^ T-cell proliferation and IFN-γ production. Building on this foundation, the present study investigates the effects and underlying mechanisms of CPP-3a on RAW264.7 macrophages, with a particular focus on its role in promoting M1 polarization. Through an examination of the immunomodulatory effects and molecular pathways associated with CPP-3a, this research aims to elucidate its potential therapeutic applications in the treatment of inflammatory and infectious diseases.

## 2. Results

### 2.1. Chlorella polysaccharide CPP-3a Induces M1 Polarization of RAW264.7 Cells

The effects of CPP-3a on RAW264.7 cells were evaluated at various time intervals by analyzing inflammatory nitric oxide (NO) secretion, cytokine production (TNF-α and IL-6), phagocytic activity, and the expression of the co-stimulatory molecule CD86. As illustrated in Figure 1, NO secretion increased starting at 6 h post-stimulation, peaking at 24 h, and then declined (Figure 1A). Similarly, TNF-α and IL-6 secretion increased significantly after 6 h, rising continuously until 48 h (Figure 1B,C). Phagocytic activity began to increase at 24 h post-stimulation and was sustained until 48 h (Figure 1D,F). The expression of the co-stimulatory molecule CD86 exhibited a significant increase 24 h after stimulation (Figure 1E,F). Collectively, enhanced secretion of pro-inflammatory cytokines (TNF-α, IL-6), elevated NO production, increased phagocytic activity, and upregulated CD86 expression demonstrate that CPP-3a induces M1 polarization in RAW264.7 cells.

### 2.2. Both TLR4 Inhibitor TAK242 and TLR2 Inhibitor C29 Inhibited CPP-3a-Induced M1 Polarization in RAW264.7 Cells

Toll-like receptors, particularly TLR4 and TLR2, are well-recognized as primary targets for polysaccharides. Consequently, we postulated that TLR4 and TLR2 might serve as targets for CPP-3a. To evaluate this hypothesis, we employed the specific TLR4 inhibitor TAK242 and the TLR2 inhibitor C29. As shown in Figure 2, pretreatment with TAK-242 completely inhibited CPP-3a-stimulated IL-6 release (Figure 2A) and NO production (Figure 2E), while significantly attenuating TNF-α secretion (Figure 2B), phagocytic enhancement (Figure 2C), and CD86 upregulation (Figure 2D). Notably, the TLR2 inhibitor C29 also significantly suppressed CPP-3a-induced changes in these parameters, though its inhibitory effects were consistently weaker than those of TAK-242 across all assays (Figure 2A–E). These results indicate that TLR4 likely serves as the primary receptor for CPP-3a, with TLR2 acting as a secondary mediator.

### 2.3. TLR4-KO Inhibits CPP-3a-Induced M1 Polarization in RAW264.7 Cells

To definitively establish TLR4 as the molecular target of CPP-3a, TLR4-knockout RAW264.7 cells were generated via CRISPR-Cas9. Following sequential puromycin selection and limiting dilution, DNA sequencing revealed a homogeneous adenine deletion at position 2046. Bioinformatics confirmed this frameshift mutation ablates residues 682–835 within the essential TIR domain, disrupting TLR4 signaling competence (Figure 3A). Functional validation showed complete ablation of LPS responses: TLR4-KO cells maintained basal p-p65/p65 and p-p38/p38 ratios (Figure 3B,C), and displayed no LPS-induced increases in NO production, IL-6/TNF-α secretion, CD86 upregulation, or phagocytosis (Figure 3D–H). When challenged with CPP-3a, these cells showed complete abrogation of NO production (Figure 3D), IL-6 release (Figure 3E), phagocytic activity (Figure 3G), and CD86 expression (Figure 3H), whereas TNF-α secretion showed partial attenuation (Figure 3F). This receptor-specific inhibition pattern confirms TLR4 as the primary mediator of CPP-3a-induced polarization, with residual TNF-α implicating ancillary pathways.

### 2.4. C29 Suppresses TNF-α Release in CPP-3a-Activated TLR4-KO RAW264.7 Macrophages

Building on our observation that the TLR2 inhibitor C29 partially attenuated CPP-3a-induced M1 polarization (Figure 2A–E), we further investigated TLR2’s role using TLR4-knockout RAW264.7 cells. Treatment with C29 significantly enhanced the inhibition of TNF-α secretion in CPP-3a-stimulated TLR4-KO cells (Figure 4A).

Additionally, Western blot analysis revealed that C29 pretreatment substantially reduced CPP-3a-triggered phosphorylation of p65 but not p38 in TLR4-KO cells (Figure 4B–D). These results establish TLR2 as a secondary signaling receptor for CPP-3a, operating independently of TLR4 to mediate residual TNF-α production and partial NF-κB activation.

### 2.5. CPP-3a Activates the TLR Signaling Pathway Through MyD88

Upon activation of Toll-like receptors, signaling can proceed through both MyD88-dependent and MyD88-independent pathways. To elucidate the specific signaling pathways activated by CPP-3a, we assessed the protein expression levels of MyD88 and TRIF in RAW264.7 cells at 24 h post-stimulation with CPP-3a. The findings, presented in Figure 5, demonstrate a significant upregulation of MyD88 protein levels following CPP-3a stimulation, whereas TRIF expression was undetectable.

### 2.6. The NF-κB and p38 MAPK Signaling Are Activated in CPP-3a-Stimulated RAW264.7 Cells

Complementary investigations established that CPP-3a activates NF-κB and MAPK signaling to drive M1 polarization in RAW264.7 cells. As shown in Figure 6A–C, CPP-3a stimulation significantly enhanced phosphorylation of NF-κB p65 and facilitated its nuclear translocation. Parallel activation of the MAPK pathways was evidenced by increased phosphorylation of JNK, ERK1/2, and p38 (Figure 6D–G). To determine pathway-specific contributions to the polarization phenotype, we employed pharmacological inhibitors targeting these cascades. Pretreatment with an NF-κB inhibitor substantially suppressed CPP-3a-induced phagocytosis enhancement, NO secretion, CD86 upregulation, and IL-6 production (Figure 7A–D). Among MAPK inhibitors, the p38 blocker partially attenuated phagocytic potentiation, NO release, and CD86 expression, whereas the JNK inhibitor only modestly reduced IL-6 secretion. Notably, MEK1/2 inhibition (targeting ERK1/2 upstream due to the lack of direct ERK inhibitors) showed no significant effects (Figure 7A–D). These differential inhibition profiles demonstrate that CPP-3a primarily induces M1 polarization through coordinated activation of NF-κB and p38 signaling pathways.

## 3. Discussion

Our findings establish that *Chlorella* polysaccharide CPP-3a robustly induces M1 polarization in RAW264.7 macrophages. This conclusion is grounded in comprehensive phenotypic evidence: enhanced secretion of inflammatory mediators (NO, TNF-α, and IL-6; Figure 1A–C), significant upregulation of phagocytic activity (Figure 1D), and elevated expression of the co-stimulatory molecule CD86 (Figure 1E). These functional changes collectively position CPP-3a as a potent innate immune activator, capable of priming macrophages for pathogen defense and inflammatory responses—a property with potential implications for vaccine adjuvants or immunotherapies.

Critically, we identified TLR4 as the dominant receptor mediating CPP-3a immunomodulatory effects. Unlike prior studies limited to single-method validation like using a TLR4-specific inhibitor [21,22,23] or TLR4-blocking antibodies [13,24,25,26], we employed a dual-approach strategy combining pharmacological inhibition using TAK242 (Figure 2) with genetic ablation via CRISPR-Cas9-generated TLR4-KO RAW264.7 cells (Figure 3), both approaches abolished CPP-3a-induced M1 polarization markers; this orthogonal validation conclusively establishes TLR4 as the primary target of CPP-3a. Notably, residual TNF-α secretion in TLR4-KO cells (Figure 3G) implies additional receptor involvement, prompting further investigation.

Building upon established evidence that TLR2 serves as a recognition receptor for immunomodulatory polysaccharides [13,27,28,29], our pharmacological inhibition assays using the TLR2-specific antagonist C29 further substantiate its role as a secondary target for CPP-3a (Figure 4). Critically, however, neither TLR4 nor TLR2 blockade completely abrogated TNF-α secretion (Figure 3F and Figure 4A), implying compensatory signaling via alternative receptors such as complement receptor 3 [30,31,32] and dectin-1 [33], which can activate parallel NF-κB pathways to sustain cytokine production and inflammatory responses even under TLR suppression. Future studies should resolve these multi-receptor dynamics through structural mapping of CPP-3a-binding interfaces and conditional knockout models.

The significant activation of the NF-κB and p38 MAPK pathways, as demonstrated by elevated phosphorylation levels (Figure 6A–B,D–G), nuclear translocation of NF-κB subunits (Figure 6C), and significant suppression by selective inhibitors (Figure 7)—confirms their central role in CPP-3a-induced immune modulation. These findings are consistent with some previously reported studies on the pathways of polysaccharides stimulating macrophage activation [34,35,36,37,38], and also align well with our recent observations in DCs, where CPP-3a induced maturation by enhancing similar co-stimulatory molecules and cytokine production [6]. The conserved activation pattern across macrophages and DCs suggests a shared receptor-initiated mechanism, potentially pivotal for understanding how *Chlorella* polysaccharides exert pan-immune effects.

The dual immunomodulatory capacity of CPP-3a to drive M1 macrophage polarization and human dendritic cell maturation underscores its therapeutic promise for human pathologies demanding amplified innate immunity. Critically, this functional synergy is mediated through the evolutionarily conserved TLR4/MyD88/NF-κB axis, which exhibits >80% homology between murine and human systems [15]. Such mechanistic conservation positions CPP-3a as a prime candidate for use as a human vaccine adjuvant. Unlike synthetic adjuvants (e.g., alum, MF59) that often provoke localized inflammation [39], CPP-3a’s natural origin and structural biocompatibility may mitigate systemic adverse effects while maintaining robust immunostimulatory potency. Future studies focused on the integration of CPP-3a in vaccine formulations and subsequent in vivo trials will be crucial to ascertain its effectiveness and safety as an adjuvant.

Studies have shown that the molecular weight, protein and sulfate content, and monosaccharide composition of *Chlorella* polysaccharides influence their immunomodulatory activity [3,20]. The complexity of polysaccharide structures plays a crucial role in their biological activities. For instance, β-glucans, a well-studied form of polysaccharides, exhibit varying immunological effects based on their size and branching patterns [40,41,42]. Our previous structural analysis revealed that CPP-3a has a complex structure: →2)-α-L-Araf-(1→, →2)-α-D-Rhap-(1→, →5)-α-L-Araf-(1→, →3)-β-D-Glcp-(1→, →4)-α-D-Glcp-(1→, →4)-α-D-GlcpA-(1→, →2,3)-α-D-Manp-(1→, →3,4)-α-D-Manp-(1→, →3,4)-β-D-Galp-(1→, →3,6)-β-D-Galp-(1→, and →2,3,6)-α-D-Galp-(1→ residues, with branches at C2, C3, C4, or C6 of α/β-D-Galp and α-D-Manp, and terminal groups of α/β-L-Araf, α-L-Arap, α-D-Galp, and β-D-Glcp. This complexity may explain why CPP-3a exerts multi-cellular, multi-receptor, and multi-pathway regulatory effects. Moreover, previous studies have highlighted that the immunological activity of *Chlorella* polysaccharides is largely dependent on their monosaccharide composition, which predominantly includes Gal, Rha, and Ara [18]. Among TLR4-associated polysaccharides, Glc, Gal, and Man are the most frequently encountered monosaccharides [43]. CPP-3a is rich in these key monosaccharides, including Glc, Gal, Ara, Rha, and Man. These compositional features likely confer CPP-3a with the capacity to promote M1 polarization in RAW264.7 cells, primarily via its interaction with TLR4. Further detailed studies on the structure-activity relationship are needed to elucidate these mechanisms.

While this study systematically delineates CPP-3a’s TLR4/MyD88-dependent mechanism driving M1 polarization in RAW264.7 macrophages, three strategic limitations warrant emphasis: firstly, the exclusive reliance on an immortalized murine cell line necessitates future corroboration in primary macrophage models—such as human monocyte-derived macrophages or murine bone marrow-derived macrophages—to bridge in vitro findings with physiological contexts; secondly, although canonical M1 markers were comprehensively assessed, motility-associated phenotypes including chemotactic responsiveness and transendothelial migration capacity remain uncharacterized—an aspect prioritized in forthcoming investigations; thirdly, while our data confirm no short-term cytotoxicity of CPP-3a in RAW264.7 cells (Appendix A), quantitative assessment of CPP-3a’s long-term biocompatibility—specifically sustained polarization stability and systemic toxicity risks—remains imperative to de-risk therapeutic translation.

## 4. Materials and Methods

### 4.1. Reagents

CPP-3a was isolated and purified from dried *Chlorella pyrenoidosa* powder according to previously established methods [4]. The endotoxin level in CPP-3a was verified to be less than 0.05 EU/mg via LAL assay as described in our previous study [6]. RAW264.7 cells were obtained from Procell (Wuhan, China, #CL-0190). Dulbecco’s Modified Eagle Medium (DMEM) high glucose, supplemented with 10% fetal bovine serum (FBS), was acquired from Gibco (Waltham, MA, USA). RAW264.7 cells were maintained through gentle mechanical dissociation without enzymatic digestion: at 70–80% confluence, adherent cells were detached using sterile tips in culture medium, followed by centrifugation at 300× *g* for 5 min. All experiments utilized cells within passages 5–15, with split ratios of 1:3 to 1:10 every 48–72 h. Lipofectamine 2000 was from Invitrogen (Carlsbad, CA, USA, #2462811). Lipopolysaccharide (LPS-EB) was purchased from InvivoGen (San Diego, CA, USA, #tlrl-eblps). Griess reagent was sourced from Solarbio (Beijing, China, #S0021S). TAK242, C29, U0126, Adezmapimod, SP600125, Pyrrolidinedithiocarbamate ammonium, and Fluorescein isothiocyanate (FITC)-dextran (Mw 10,000) were obtained from MedChemExpress (Shanghai, China, #HY-11109, #HY-100461, #HY-12031A, #HY-10256, #HY-12041, #HY-18738, #HY-128868, respectively). Human TruStain FcX was purchased from BioLegend (San Diego, CA, USA, #564219), and PE-conjugated rat anti-mouse CD86 (clone GL1) was purchased from BD Biosciences (Franklin Lakes, NJ, USA, #561963). Antibodies for GAPDH, β-actin, coraLite 488-conjugated goat anti-rabbit IgG (H+L), and ELISA kits for Human TNF-α and IL-6 were acquired from Proteintech (Wuhan, China, #10494-1-AP, #66009-1-Ig, #SA00013-2, #KE10002, #KE10007). Antibodies for NF-κB/p65, phosphorylated NF-κB/p65, JNK1/2, and phosphorylated JNK1/2 were sourced from Cell Signaling Technology (Beverly, MA, USA, #8242T, #3033, #9252T, #4668T). Antibodies for ERK1/2, phosphorylated ERK1/2, p38 MAPK, phosphorylated p38 MAPK, and phosphorylated JNK1/2 were from Proteintech (Wuhan, China, #83533-1, #80031-1, #80821-3, #28796-1, #80024-1). MyD88 (V220) polyclonal antibody was purchased from Bioworld (Nanjing, China, #BS3521). sgRNAs were synthesized by Sangon Biotech (Shanghai, China). The genomic DNA purification kit was from Thermo Fisher Scientific (Waltham, MA, USA, #K0712). 2X SanTaq PCR Master Mix was purchased from Sangon Biotech (Shanghai, China, #B532061-0040). Cell Counting Kit-8 was from Beyotime (Shanghai, China, #C0038). All the other chemicals and reagents used were of analytical grade.

### 4.2. Phagocytic Function Assay

Phagocytic activity was assessed using FITC-labeled dextran. A sample of 1 × 10^5^ cells in 100 µL of DMEM was incubated with 1 mg/mL FITC-dextran in a 5% CO_2_ humidified incubator at 37 °C for 30 min. Cells were washed twice with PBS containing 0.1% BSA and then fixed in 1% paraformaldehyde (PFA). The uptake of FITC-dextran by the cells was quantified using standard flow cytometry techniques.

### 4.3. Assessment of Nitrite Levels and Cytokine Release

RAW264.7 cells were seeded at a density of 1 × 10^5^ cells/well in 24-well plates and cultured for 24 h. Cells were then treated with specified concentrations of CPP-3a or LPS (1 µg/mL). At designated time points, culture media were collected for nitrite and cytokine quantification using the Griess reagent and ELISA kits, respectively, following the manufacturer’s instructions.

### 4.4. Flow Cytometry Analysis

RAW264.7 cells were harvested at specified time points after treatment, washed twice with PBS containing 0.1% BSA at room temperature, and resuspended in 100 µL of staining buffer. Cells (1 × 10^5^) were blocked with 5 µL of Human TruStain FcX™ at 4 °C for 15 min, followed by incubation with PE-conjugated goat anti-mouse CD86 in the dark at 4 °C for 30 min. After post-incubation, cells were washed and fixed in 4% PFA before analysis by flow cytometry.

### 4.5. Western Blotting

RAW264.7 cells were plated at a density of 2 × 10^5^ cells/mL per well in 24-well plates and cultured overnight. After treatment for specified periods, total cellular proteins were extracted using RIPA buffer supplemented with 1% PMSF and 1% protease inhibitor cocktail. Protein concentrations were quantified using a BCA Protein Assay Kit (Epizyme Biotech, Shanghai, China). Proteins were denatured and separated on 12% SDS-PAGE gels, then transferred onto PVDF membranes (Millipore, Billerica, MA, USA). The membranes were blocked with Protein Free Rapid Blocking Buffer (Epizyme Biotech, Shanghai, China) for 10 min, followed by overnight incubation at 4 °C with primary antibodies diluted 1:1000. After washing with TBST, the membranes were incubated with horseradish peroxidase-conjugated secondary antibodies at a 1:3000 dilution for 1 h at room temperature. Following further washes, protein bands were visualized using a CHAMPCHEMI Imaging System (Sagecreation, Peiking, China) or Touch Imager (e-blot, Shanghai, China). GAPDH or β-actin served as a loading control.

### 4.6. Establishment of Tlr4 Knockout Cell Line

Three single-guide RNAs (sg-RNAs) targeting the gene sequence of mouse *Tlr4* were designed using the CHOPCHOP online tool (http://chopchop.cbu.uib.no/, [accessed on 1 March 2023]), synthesized, and subsequently cloned into the lentiCRISPRv2 vector (refer to Table 1). The resulting plasmids, along with empty vectors, were transformed into *Stbl3* cells.

Following selection, the expanded clones were utilized for plasmid extraction. These cloned plasmids, in conjunction with the packaging plasmids psPAX2 and pVSVg at a ratio of 5:4:1, were co-transfected into HEK293FT cells using Lipofectamine 2000 reagent to produce lentivirus. This lentivirus was subsequently employed to infect RAW264.7 cells. Following infection, the cells underwent selection with puromycin and were then cloned using the limiting dilution method to establish stable gene knockout cell lines. The purified genomic DNA from these cells was used to amplify the gene editing region, and the resulting PCR products were sequenced to verify the success of the gene editing process.

### 4.7. Immunofluorescence Assay

RAW264.7 cells were seeded at a density of 1 × 10^5^ cells/mL in confocal dishes overnight. Subsequently, cells were treated with specified concentrations of CPP-3a or LPS (1 µg/mL) for 2 h. Cells were then fixed, permeabilized, and blocked before overnight incubation with NF-κB p65 antibody (1:100) at 4 °C. Afterward, cells were incubated with CoraLite 488-conjugated goat anti-rabbit IgG (H+L) for 1 h. Nuclei were stained with DAPI. Fluorescence images were captured using an Axio Observer 7 equipped with Apotome 2 (ZEISS, Oberkochen, Germany).

### 4.8. Statistical Analysis

Data are presented as mean ± standard error of the mean (SEM). Statistical analyses were performed using One-way ANOVA or *t*-tests with GraphPad Prism 8 software. A *p*-value of less than 0.05 was considered statistically significant. Significance levels are indicated as follows: * *p* < 0.05, ** *p* < 0.01, *** *p* < 0.001, **** *p* < 0.0001.

## 5. Conclusions

This study reveals the significant immunomodulatory effects of *Chlorella pyrenoidosa* polysaccharide CPP-3a on macrophage polarization, underscoring its potential therapeutic applications. The identification of TLR4 and TLR2 as key receptors suggests a receptor-mediated mechanism for the effects of CPP-3a. By elucidating its influence on the NF-κB and p38 MAPK signaling pathways, the research provides essential insights into the molecular mechanisms underlying the actions of CPP-3a. These findings suggest CPP-3a as a promising candidate for the management of inflammatory and infectious diseases. Future research will focus on further elucidating the complex receptor-ligand interactions and evaluating its efficacy and safety as a vaccine adjuvant in in vivo trials.

## Figures and Tables

**Figure 1 marinedrugs-23-00290-f001:**
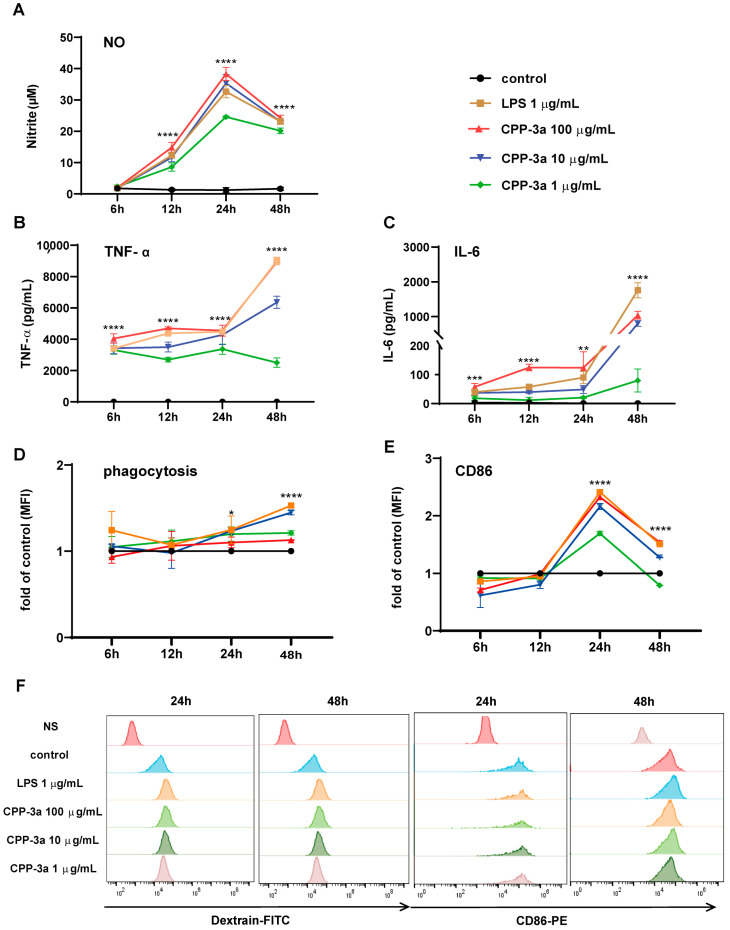
Effect of CPP-3a on the secretion of NO (**A**), TNF-α (**B**), IL-6 (**C**), phagocytic function (**D**), and CD86 expression (**E**) of RAW264.7 cells. A representative result of phagocytic function and CD86 expression (**F**) is presented. Significance levels are indicated as follows: * *p* < 0.05, ** *p* < 0.01, *** *p* < 0.001, **** *p* < 0.0001.

**Figure 2 marinedrugs-23-00290-f002:**
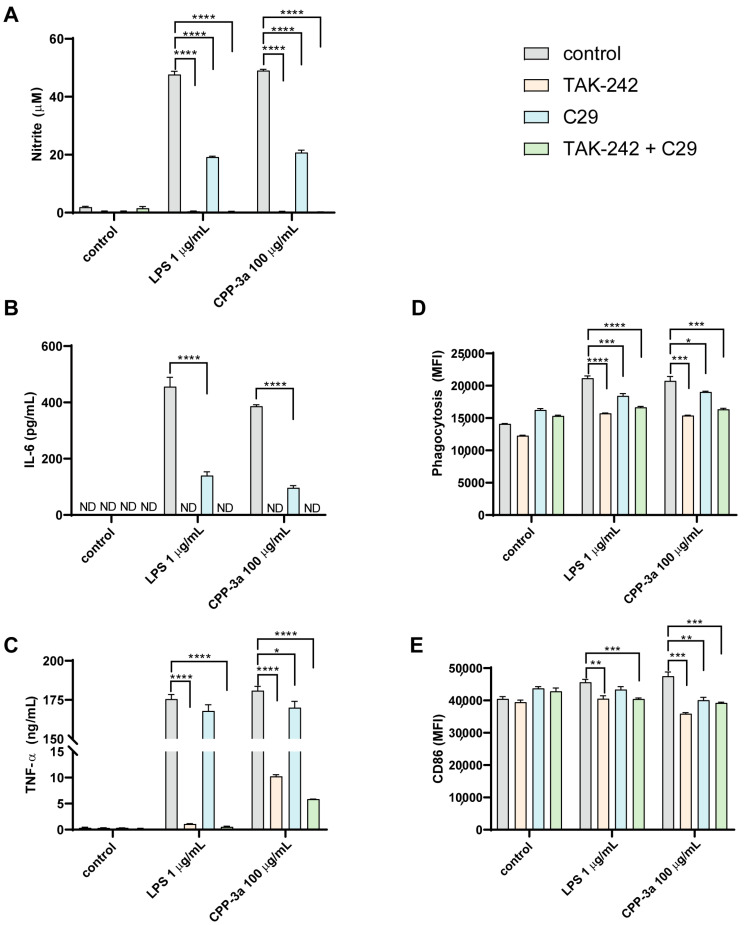
The effects of TLR4-specific inhibitor TAK-242 and TLR2-specific inhibitor C29 on CPP-3a-stimulated NO production (**A**), IL-6 (**B**), TNF-α secretion (**C**), phagocytosis (**D**), and CD86 expression (**E**) in RAW264.7 cells. Significance levels are indicated as follows: * *p* < 0.05, ** *p* < 0.01, *** *p* < 0.001, **** *p* < 0.0001.

**Figure 3 marinedrugs-23-00290-f003:**
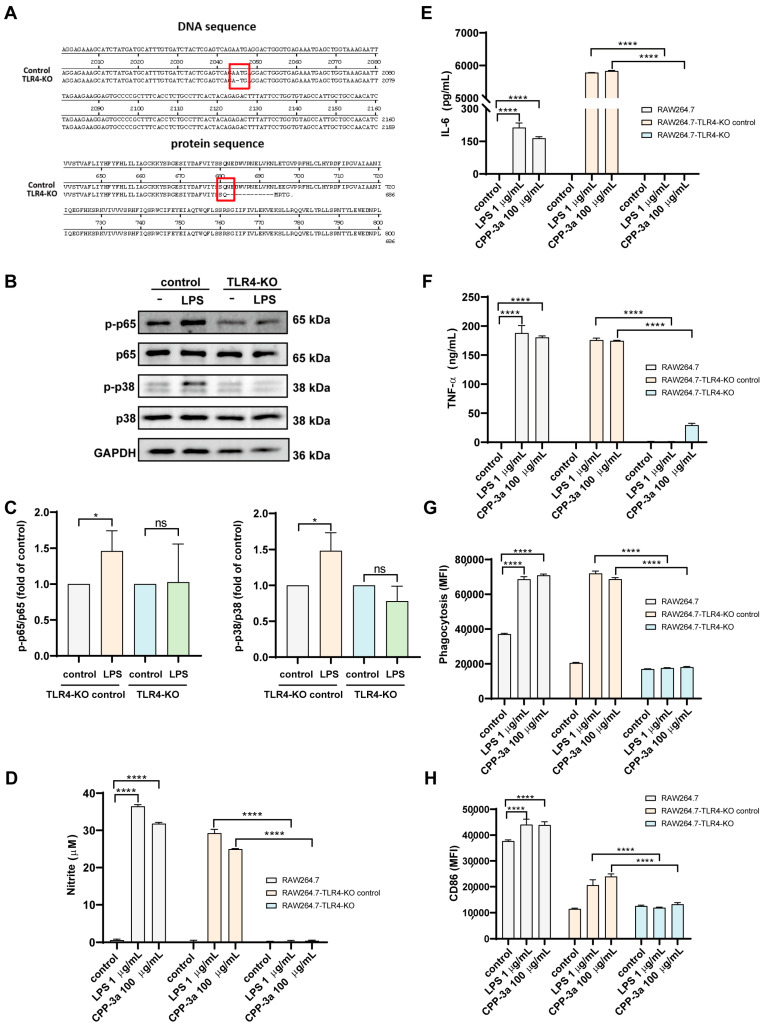
Comparison of DNA sequences and the protein sequence of the control cells and the *Tlr4* gene-knockout cells (**A**). Comparison of the expression of phosphorylated-p65 and -p38 in the control cells and *Tlr4* gene-knockout cells under LPS stimulation (**B**,**C**). Comparison of NO (**D**), IL-6 (**E**), TNF-α (**F**) production, phagocytosis (**G**), and CD86 expression (**H**) between control cells and *Tlr4* gene-knockout cells stimulated with LPS or CPP-3a. Significance levels are indicated as follows: * *p* < 0.05, **** *p* < 0.0001, ns denotes not significant.

**Figure 4 marinedrugs-23-00290-f004:**
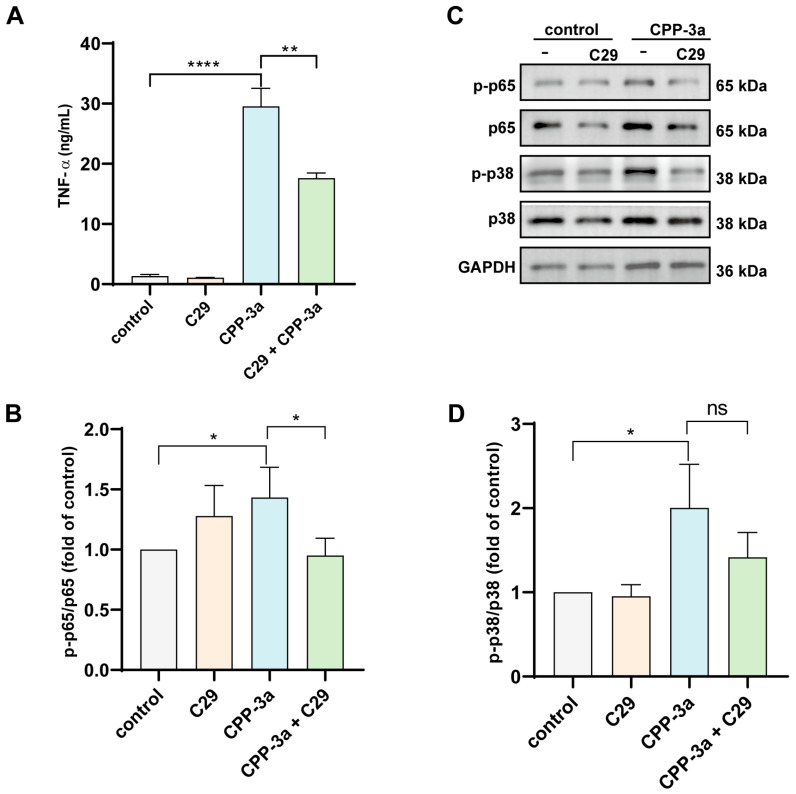
Effect of TLR2-specific inhibitor C29 on TNF-α (**A**) production in TLR4-KO cells stimulated with CPP-3a. Effect of C29 on the expression of phosphorylated-p65 (**B**,**C**) and -p38 (**C**,**D**) in TLR4-KO cells stimulated with CPP-3a. Significance levels are indicated as follows: * *p* < 0.05, ** *p* < 0.01, **** *p* < 0.0001, ns denotes not significant.

**Figure 5 marinedrugs-23-00290-f005:**
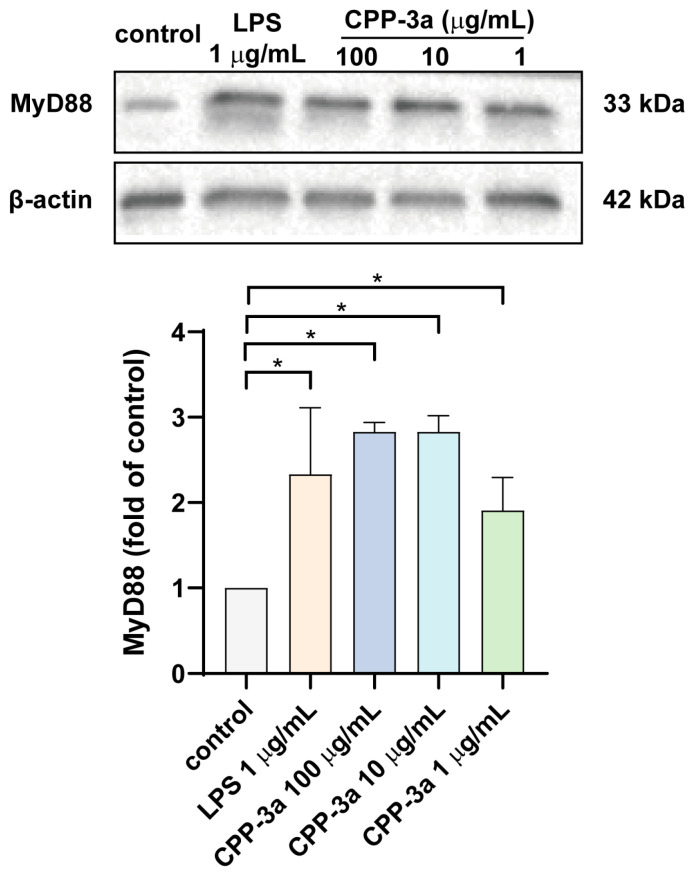
Expression of MyD88 in CPP-3a-stimulated RAW264.7 cells. Significance levels are indicated as follows: * *p* < 0.05.

**Figure 6 marinedrugs-23-00290-f006:**
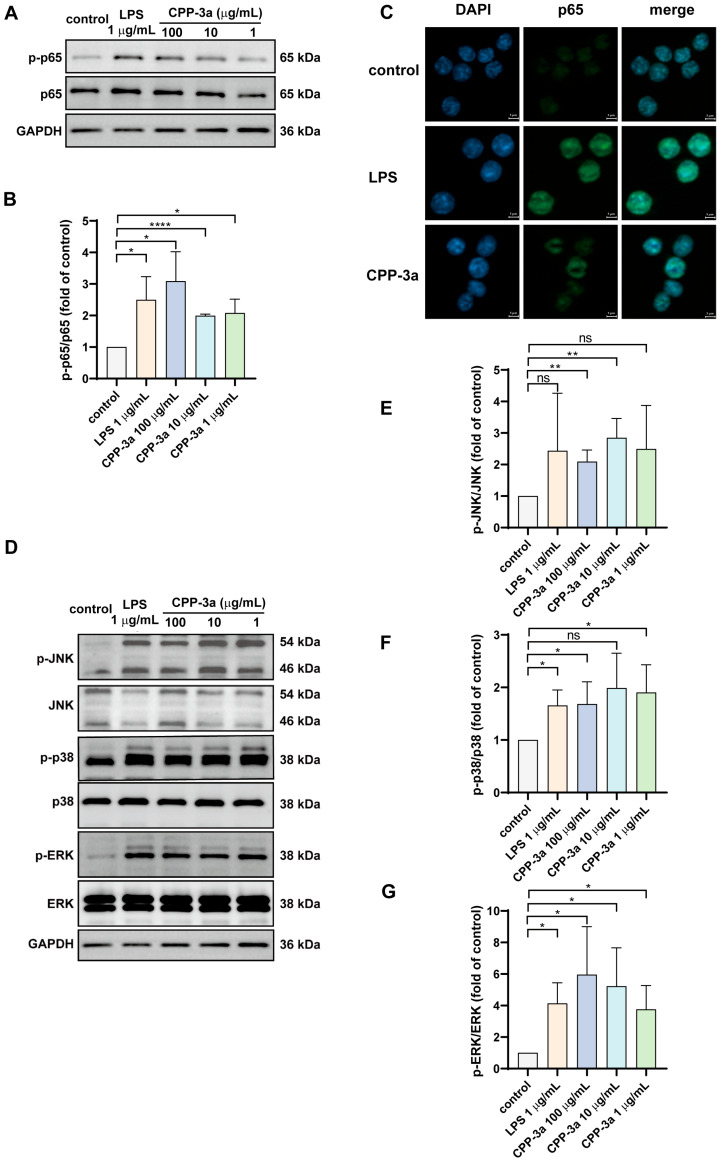
Effect of CPP-3a on the expression of phosphorylated p65 (**A**,**B**) and its nuclear translocation (**C**) in RAW264.7 cells. CPP-3a enhanced the expression of phosphorylated JNK (**D**,**E**), p38 (**D**,**F**), and ERK (**D**,**G**). Significance levels are indicated as follows: * *p* < 0.05, ** *p* < 0.01, **** *p* < 0.0001, ns denotes not significant.

**Figure 7 marinedrugs-23-00290-f007:**
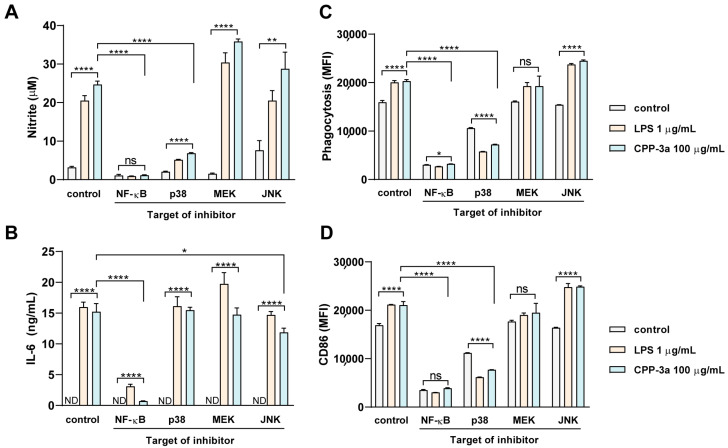
Effect of specific inhibitors on CPP-3a-stimulated NO (**A**) and IL-6 secretion (**B**), phagocytosis (**C**), and CD86 expression (**D**) in RAW264.7 cells. Significance levels are indicated as follows: * *p* < 0.05, ** *p* < 0.01, **** *p* < 0.0001, ns denotes not significant.

**Table 1 marinedrugs-23-00290-t001:** sgRNA design for mouse *Tlr4* gene knock-out.

	Target Sequence	Synthesized Oligo DNA	Primers for Verification
Sg1	5’-ACACGTCCATCGGTTGATCTTGG-3’	5’-CACCGACACGTCCATCGGTTGATCT-3’ 3’-CTGTGCAGGTAGCCAACTAGACAAA-5’	Left primer: GGGAATTAAGCTCCATGAACTG Right primer: GATACACCTGCCAGAGACATTG
Sg2	5’- GATCTACTCGAGTCAGAATGAGG -3’	5’-CACCGGATCTACTCGAGTCAGAATG-3’ 3’-CCTAGATGAGCTCAGTCTTACCAAA-5’	Left primer: TCATCAGTGTGTCAGTGGTCAG Right primer: TGTAGTGAAGGCAGAGGTGAAA
Sg3	5’- CACGTCCATCGGTTGATCTTGGG -3’	5’-CACCGCACGTCCATCGGTTGATCTT-3’ 3’-CGTGCAGGTAGCCAACTAGAACAAA-5’	Left primer: GGGAATTAAGCTCCATGAACTG Right primer: GATACACCTGCCAGAGACATTG

## Data Availability

The authors declare that the data supporting the findings of this study are presented in the article.

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
