# Peer review of "Chlorella pyrenoidosa Polysaccharide CPP-3a Promotes M1 Polarization of Macrophages via TLR4/2-MyD88-NF-κB/p38 MAPK Signaling Pathways"

_marinedrugs, 2025, doi:10.3390/md23070290_

Round 1

Reviewer 1 Report

Comments and Suggestions for Authors

In this manuscript, the authors demonstrated that purified Chlorella pyrenoidosa polysaccharide (CPP-3a), a derivative of Chlorella pyrenoidosa microalga, can affect immune cell behavior. The authors showed that, similarly to LPS, CPP-3a promoted RAW264.7 macrophage IL-6 and TNF-a cytokine secretion, nitric oxide, phagocytosis and differentiation.  They demonstrated that the CPP3a-induced effects are mainly induced by TLR4 signaling using TLR4 inhibitor, and partially by TRL2 inhibitor. Nevertheless, the combination of TLR2/TLR4 inhibitors did not promote the induction. They confirmed via lack of function assay that TLR4 is responsible of CPP3a-inudced effects, suggesting the preponderance of TLR4 signaling. They also demonstrated that the response is through My88-dependant pathway. Moreover, they demonstrated that CPP-3a activated NF-κB and p38 MAPK signaling using specific inhibitors. Altogether, the data suggested that CPP-3a induces M1 polarization in RAW264.7 macrophages. Globally, this work is clear and convincing. Nevertheless, there are few comments to consider before that the manuscript may be considered for publication in Marine Drugs.

Major comments:

  • The use of an alternative cell line or primary cell (in particular human one) for the key experiments would be valuable for this manuscript to confirm the data obtained with murine RAW264.7 cells.
  • The authors did not evaluate the effect of CPP3a on cell viability.
  • Functional analysis of the M1 polarization would be valuable to this work by analyzing for example cell motility.

Minor comments:

  • The authors are asked to indicate numbers of passages of the cell lines used and the original passage from biobank’s provider if relevant. The method to sub cultivate cells is not described.
  • The authors did not describe the potential interest of CPP-3a for human research. Additional details in the discussion will help to emphasize the interest of this research for the scientific community.

Reviewer 2 Report

Comments and Suggestions for Authors

In this study, authors tried to investigate the effects and underlying mechanisms of Chlorella pyrenoidosa polysaccharide CPP-3a on RAW264.7 macrophages. In my opinion, the experimental design of this paper is solid and fulfills a very sufficient amount of work. However, there are some issues with the writing of this paper and some details that I would like the author to revise.

  1. Line269-270,NF-κB, not NF-B.
  2. Table 1 is not in a standard table format.
  3. Some content appears to be in Chinese font, such as the "κ" in NF-κB under Abbreviations. There are many such errors in the manuscript, including in the figures. The author is advised to check the whole paper carefully.
  4. Line 241-251, the discussion in this section is weak and the relationship between structure and function of polysaccharides in different species should be described in particular.
  5. The μ font of Figure 5 also looks weird.
  6. There are 3 co-corresponding authors on this paper, and I don't know if marine drugs' policy allows for that, because as I know, some journals in MDPI have limits on the number of co-corresponding authors.
  7. Line 67-75, The author should have been more specific and detailed about their previous work, especially CPP-3a.
  8. Figure 1F and Figure 1G can be combined.

Reviewer 3 Report

Comments and Suggestions for Authors

The study examines the immunomodulatory effect of a well-characterized Chlorella pyrenoidosa polysaccharide (CPP-3a) to make a valuable contribution to macrophage polarization. The authors provide a systematic mechanistic study, combining pharmacological inhibitors and CRISPR-Cas9-induced knockout of TLR4 to show conclusively that TLR4 is the predominant receptor involved in CPP-3a-induced M1 polarization and TLR2 as the second receptor.
Comments for author
1. Here, in vitro experiments, I observe in this article that all experiments are conducted in RAW264.7 cell lines. In the absence of in vivo validation or use of primary macrophages, direct applicability to clinical or physiological conditions is limited. I would suggest that the author include in vivo studies or primary cell experiments to add translational relevance.
2. In regards to structural-functional correlation within the current study authors do not correlate specific structural motifs to functional outcomes directly, potentially optimizing mechanistic understanding.
3. In the case of Cytokine Analysis experiments author focuses only on TNF-α and IL-6. It is my suggestion to the author that a full cytokine profiling and maker profiling (e.g., IL-12, IL-1β, anti-inflammatory markers) would provide a better characterization of the polarization spectrum and rule out mixed or aberrant activation states.
4. One of the important perspectives in this study is not mentioned and that is endotoxin contamination. Endotoxin contaminants are potent agonists of TLR4, and microalgal polysaccharide preparations can be contaminated with them. I would suggest that the author address possible endotoxin contamination with appropriate controls (e.g., polymyxin B treatment, LAL assay).
5. The author of this research does not mention the long-term outcome or safety of CPP-3a-induced macrophage activation, which is important for therapeutic use.
Discussion of the mentioned limitations would further increase the impact and translational significance of this potential study.

Round 2

Reviewer 1 Report

Comments and Suggestions for Authors

The authors provided relevant answers to the reviewer's comments. Once again, this work is well-designed, and I am quite interested in seeing the authors' further work using primary cells. This work could be published.

Reviewer 3 Report

Comments and Suggestions for Authors

I have thoroughly evaluated the revised manuscript, along with the authors' comprehensive responses to the previous comments and suggestions. The authors have adequately addressed most concerns and given clarifications throughout the manuscript. Accordingly, I recommend to the editor regarding this article to make a decision to accept for publication in this esteemed journal.